# GradiVeQ: Vector Quantization for Bandwidth-Efficient Gradient Aggregation in Distributed CNN Training

Mingchao Yu$^{\diamond\ast}$, Zhifeng Lin$^{\diamond\ast}$, Krishna Narra$^{\diamond}$, Songze Li$^{\diamond}$, Youjie Li$^{\dagger}$, Nam Sung Kim$^{\dagger}$,
Alexander Schwing$^{\dagger}$, Murali Annavaram$^{\diamond}$, and Salman Avestimehr$^{\diamond}$

$^{\diamond}$University of Southern California
$^{\dagger}$University of Illinois at Urbana Champaign

## Abstract

Data parallelism can boost the training speed of convolutional neural networks (CNN), but could suffer from significant communication costs caused by gradient aggregation. To alleviate this problem, several scalar quantization techniques have been developed to compress the gradients. But these techniques could perform poorly when used together with decentralized aggregation protocols like ring all-reduce (RAR), mainly due to their inability to directly aggregate compressed gradients. In this paper, we empirically demonstrate the strong *linear correlations* between CNN gradients, and propose a gradient *vector* quantization technique, named GradiVeQ, to exploit these correlations through principal component analysis (PCA) for substantial gradient dimension reduction. GradiVeQ enables direct aggregation of compressed gradients, hence allows us to build a distributed learning system that parallelizes GradiVeQ gradient compression and RAR communications. Extensive experiments on popular CNNs demonstrate that applying GradiVeQ slashes the wall-clock gradient aggregation time of the original RAR by more than $5X$ without noticeable accuracy loss, and reduces the end-to-end training time by almost $50\%$. The results also show that GradiVeQ is compatible with scalar quantization techniques such as QSGD (Quantized SGD), and achieves a much higher speed-up gain under the same compression ratio.

## 1   Introduction

Convolutional neural networks (CNN) such as VGG [1] and ResNet [2] can achieve unprecedented performance on many practical applications like speech recognition [3, 4], text processing [5, 6], and image classification on very large datasets like CIFAR-100 [7] and ImageNet [8]. Due to the large dataset size, CNN training is widely implemented using distributed methods such as data-parallel stochastic gradient descent (SGD) [9, 10, 11, 12, 13, 14, 15, 16], where gradients computed by distributed nodes are summed after every iteration to update the CNN model of every node[2]. However, this gradient aggregation can dramatically hinder the expansion of such systems, for it incurs significant communication costs, and will become the system bottleneck when communication is slow.

To improve gradient aggregation efficiency, two main approaches have been proposed in the literature, namely gradient compression and parallel aggregation. Gradient compression aims at reducing the

---

$^{\ast}$ M. Yu and Z. Lin contributed equally to this work.

$^{2}$ We focus on synchronized training. Our compression technique could be applied to asynchronous systems where the model at different nodes may be updated differently [17, 18, 19, 20, 21, 22]

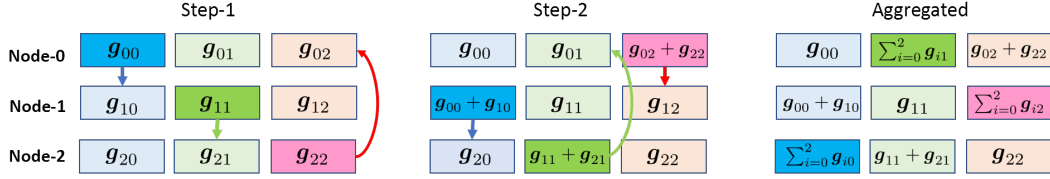

Figure 1: An example of ring all-reduce (RAR) with 3 nodes. Each node has a local vector $\boldsymbol{g}_n$. The goal is to compute $\boldsymbol{g} = \sum_{n=0}^{2} \boldsymbol{g}_n$ and share with every node. Each node will initiate the aggregation of a 1/3 segment of the vector. After 2 steps, every segment will be completely aggregated. The aggregated segments will then be simultaneously circulated to every node.

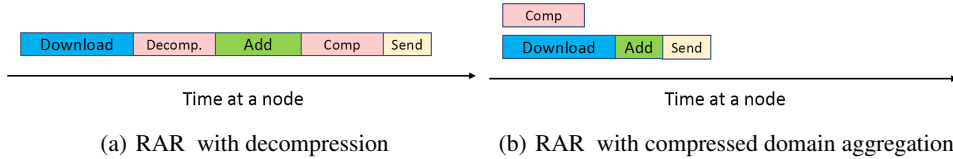

(a) RAR with decompression      (b) RAR with compressed domain aggregation

Figure 2: Processing time flow at a node in one RAR step. GradiVeQ allows the node to compress its local gradient segment while downloading, and then sum the two compressed segments and send.

number of bits used to describe the gradients. Popular methods include gradient scalar quantization (lossy) [23, 24, 25, 26, 27] and sparsity coding (lossless) [24, 28]. Parallel aggregation, on the other hand, aims at minimizing communication congestion by shifting from centralized aggregation at a parameter server to distributed methods such as ring all-reduce (RAR) [29, 30, 31, 32]. As demonstrated in Fig. 1, RAR places all nodes in a logical ring, and then circulates different segments of the gradient vector through the nodes simultaneously. Upon the reception of a segment, a node will add to it the same segment of its own, and then send the sum to the next node in the ring. Once a segment has been circulated through all nodes, it becomes a fully aggregated gradient vector. Then, another round of circulation will make it available at all nodes.

Due to the complementary nature of the above two approaches, one may naturally aim at combining them to unleash their gains simultaneously. However, there lies a critical problem: the compressed gradients cannot be directly summed without first decompressing them. For example, summing two scalar quantized gradients will incur overflow due to limited quantization levels. And, summing two sparsity coded descriptions is an undefined operation. An additional problem for sparsity based compression is that the gradient density may increase rapidly during RAR [28], which may incur exploding communication costs.

The inability to directly aggregate compressed gradients could incur hefty compression-related overheads. In every step of RAR, every node will have to decompress the downloaded gradient segment before adding its own corresponding uncompressed gradient segment. The nodes will then compress the sum and communicate it to the next node in the ring. Consequently, download and compression processes cannot be parallelized (as illustrated in Fig. 2(a)). Moreover, the same gradients will be repeatedly compressed/decompressed at every single node.

In order to leverage both gradient compression and parallel aggregation, the compression function should be *commutable* with the gradient aggregation. Mathematically, let $Q()$ be the compression function on gradient vector $\boldsymbol{g}_n$ of each node-$n$, then the following equality must hold:

$$\sum_{n=0}^{N-1} Q(\boldsymbol{g}_n) = Q\left(\sum_{n=0}^{N-1} \boldsymbol{g}_n\right),$$ (1)

where $N$ is the number of nodes. The LHS of (1) enables *compressed domain gradient aggregation*, namely, direct summation of compressed gradients. Such a compression function will allow the parallelization of compression and RAR communications, so that compression time can be masked by communication time (as illustrated in Fig. 2(b)). The RHS of (1) indicates that decompression $Q^{-1}()$ is only needed once - after the compressed gradients are fully aggregated.

**Contributions** We propose GradiVeQ (**Gradi**ent **Ve**ctor **Q**uantizer), a novel gradient compression technique that can significantly reduce the communication load in distributed CNN training. GradiVeQ is the first method that leverages both gradient compression and parallel aggregation by employing a *vector* compression technique that commutes with gradient aggregation (i.e., satisfies (1)), hence enabling *compressed domain gradient aggregation*.

**Intuition and Motivational Data:** At the core of GradiVeQ is a linear compressor that uses principal component analysis (PCA) to exploit the linear correlation between the gradients for gradient dimension reduction. Our development of GradiVeQ is rooted from the following experimental observations on the strong linear correlation between gradients in CNN:

1. **Linear correlation**: We flatten the gradients to a vector representation in a special way such that adjacent gradients could have linear correlation. As shown in Fig. 3(a), we place together the gradients located at identical coordinates across all the $F$ filters of the same convolutional layer. One of the foundational intuitions behind such a gradient vector representation is that these $F$ gradient elements are generated from the same input datum, such as a pixel value in an image, or an output from the last layer. Hence the $F$ aggregated gradients could show strong linear correlation across training iterations. This linear correlation will allow us to use PCA to compute a linear compressor for each gradient vector.

   For example, we record the value of the first 3 gradients of layer-1 of ResNet-32 for 150 iterations during CiFAR-100 training, and plot them as the 150 blue points in Fig. 4(a). As described in the figure caption, a strong linear correlation can be observed between the 3 gradients, which allows us to compress them into 2 gradients with negligible loss.

2. **Spatial domain consistency**: Another interesting observation is that, within a $(H, W, D, F)$ convolutional layer, the large gradient vector can be sliced at the granularity of $F \times D$ multiples and these slices show strong similarity in their linear correlation. This correlation is best demonstrated by the low compression loss of using the compressor of one slice to compress the other slices. For example, Fig. 5 shows that, in a $(3, 3, 16, 16)$-CNN layer, the compression loss ($\|\hat{\boldsymbol{g}} - \boldsymbol{g}\|^2 / \|\boldsymbol{g}\|^2$, where $\hat{\boldsymbol{g}}$ is the decompressed gradient vector) drops dramatically at slice sizes of 256, 512 and so on (multiple of $FD = 256$). Thus, it is possible to just perform PCA on one gradient slice and then apply the compressor to other slices, which will make PCA overhead negligible.

3. **Time domain invariance**: Finally, we also note that the observed linear correlation evolves slowly over iterations, which is likely due to the steady gradient directions and step size under reasonable learning rates. Hence, we can invest a fraction of the compute resource during a set of training iterations on uncompressed aggregations to perform PCA, and use the resulting PCA compressors to compress the successor iterations.

Built upon these observations, we develop a practical implementation of GradiVeQ, where gradient compression and RAR communications are fully parallelized. Experiments on ResNet with CIFAR-100 show that GradiVeQ can compress the gradient by 8 times, and reduce the wall-clock gradient aggregation time by over $5X$, which translates to a 46% reduction on the end-to-end training time in a system where communication contributes to 60% of the time. We also note that under the same compression ratio of 8, scalar quantizers such as 4-bit QSGD [24], while effective over baseline RAR, does not achieve the same level of performance. QSGD has the requirement that compressed gradient vectors must first be uncompressed and aggregated thereby preventing compressed domain aggregation, which in turn prevents compression and communication parallelization.

## 2   Description of GradiVeQ

In this section, we describe the core compression and aggregation techniques applied in GradiVeQ. The next section presents the details of system implementation, including system parameter selection.

We consider a distributed system that uses $N$ nodes and a certain dataset (e.g., CIFAR-100 [7] or ImageNet [8]) to train a CNN model that has $M$ parameters in its convolutional layers. We use $\boldsymbol{w}$ to represent their weights. [3]. In the $t$-th ($t \geqslant 0$) training iteration, each node-$n$ ($n \in [0, N-1]$) trains a

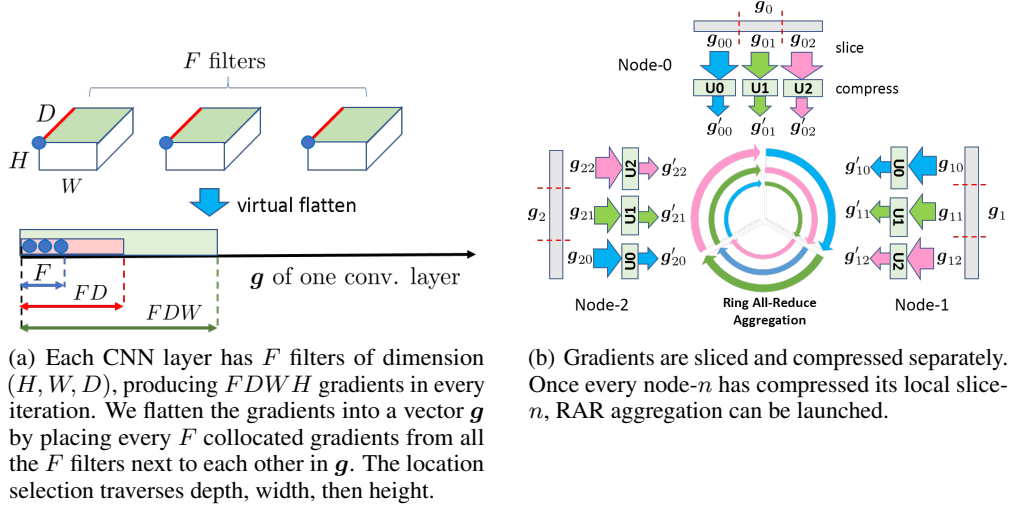

(a) Each CNN layer has $F$ filters of dimension $(H, W, D)$, producing $FDWH$ gradients in every iteration. We flatten the gradients into a vector $\boldsymbol{g}$ by placing every $F$ collocated gradients from all the $F$ filters next to each other in $\boldsymbol{g}$. The location selection traverses depth, width, then height.

(b) Gradients are sliced and compressed separately. Once every node-$n$ has compressed its local slice-$n$, RAR aggregation can be launched.

Figure 3: Gradient flattening, slicing, compression, and aggregation in GradiVeQ.

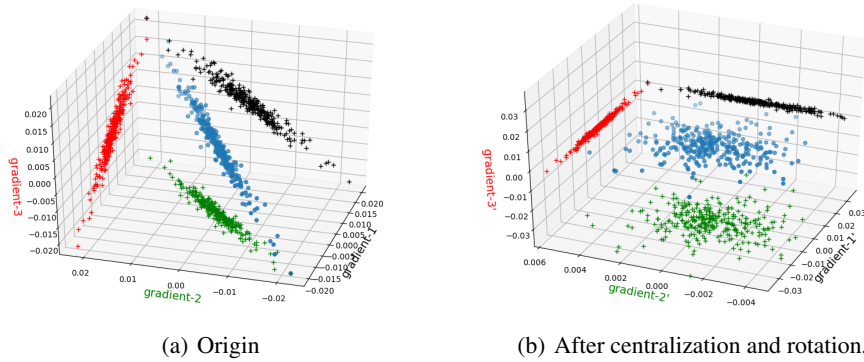

(a) Origin

(b) After centralization and rotation.

Figure 4: 3D scatter plot of the value of 3 adjacent gradients from 150 iterations (the 150 blue points in (a)), and projections to different 2D planes. A strong linear correlation is observed. After proper centralization and rotation, most of the variance/information is captured by the value of gradient $1'$ and $2'$ (the 2-D green points at the bottom plane of (b)), indicating a compression ratio of 3/2=1.5.

different subset of the dataset to compute a length-$M$ gradient vector $\boldsymbol{g}_n[t]$. These gradient vectors are aggregated into:

$$\boldsymbol{g}[t] \triangleq \sum_{n=0}^{N-1} \boldsymbol{g}_n[t], \tag{2}$$

which updates the model of every node as $\boldsymbol{w}[t+1] = \boldsymbol{w}[t] - \eta[t]\boldsymbol{g}[t]$, where $\eta[t]$ is the learning rate.

During the training phase, each convolutional layer uses the same set of filters repeatedly on sliding window over the training data. This motivates us to pack gradients $\boldsymbol{g}_n[t]$ into a vector format carefully to unleash linear correlation between adjacent gradients in $\boldsymbol{g}_n[t]$. As explained in the previous section, for a convolutional layer with $F$ filters, every $F$ collocated gradients of the $F$ filters are placed together in $\boldsymbol{g}_n[t]$. This group assignment is first applied to gradients along the filter depth, then width, and finally height (Fig. 3(a)).

GradiVeQ aims to compress every slice of $K$ adjacent gradients in $\boldsymbol{g}_n[t]$ separately, where $K$ is called the slice size. Let $\boldsymbol{g}_{n,m}[t]$ be the $m$-th ($m \in [0, M/K - 1]$) slice, GradiVeQ will compress it into $\boldsymbol{g}'_{n,m}[t]$ using a function $Q()$ as follows:

$$\boldsymbol{g}'_{n,m}[t] \triangleq Q(\boldsymbol{g}_{n,m}[t]) \triangleq \boldsymbol{U}_{d,m}^T \left( \boldsymbol{g}_{n,m}[t] - \frac{\boldsymbol{\mu}_m}{N} \right), \tag{3}$$

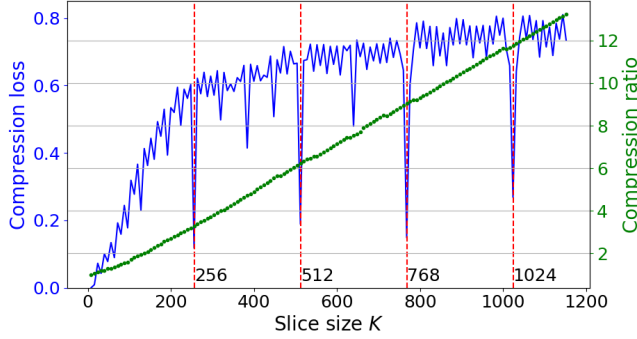

Figure 5: Slice size v.s. compression performance when using the linear compressor of the first slice of a $(3, 3, 16, 16)$-convolutional layer to compress the remaining slices. While compression ratio increases steadily with slice size, the compression loss drops drastically when slice size is a multiple of $FD = 256$, indicating similarity between the first slice's linear correlation and those of the other slices.

where $\boldsymbol{U}_{d,m}$ is a $K \times d$ linear compressor with $d < K$, and $\boldsymbol{\mu}_m$ is a length-$K$ whitening vector. After compression, the dimension of the slice is reduced to $d$, indicating a compression ratio of $r = K/d$.

The compressed slices from different nodes can then be directly aggregated into:

$$\boldsymbol{g}'_m[t] = \sum_{n=0}^{N-1} \boldsymbol{g}'_{n,m}[t] = \boldsymbol{U}_{d,m}^T \left( \boldsymbol{g}[t] - \boldsymbol{\mu}_m \right), \tag{4}$$

which indicates that GradiVeQ compression is commutable with aggregation. According to (4), a single decompression operation is applied to $\boldsymbol{g}'_m$ to obtain a lossy version $\hat{\boldsymbol{g}}_m[t]$ of $\boldsymbol{g}_m[t]$:

$$\hat{\boldsymbol{g}}_m[t] = \boldsymbol{U}_{d,m} \boldsymbol{g}'_m[t] + \boldsymbol{\mu}_m, \tag{5}$$

which will be used to update the corresponding $K$ model parameters.

In this work we rely on PCA to compute the compressor $\boldsymbol{U}_{d,m}$ and the whitening vector $\boldsymbol{\mu}_m$. More specifically, for each slice-$m$, all nodes periodically invest the same $L_t$ out of $L$ CNN training iterations on uncompressed gradient aggregation. After this, every node will have $L_t$ samples of slice-$m$, say, $\boldsymbol{g}_m[t], \cdots, \boldsymbol{g}_m[t + L_t - 1]$. The whitening vector $\boldsymbol{\mu}_m$ is simply the average of these $L_t$ samples. Every node then computes the covariance matrix $\boldsymbol{C}_m$ of the $K$ gradients using these $L_t$ samples, and applies singular value decomposition (SVD) to obtain the eigen matrix $\boldsymbol{U}_m$ and eigen vector $\boldsymbol{s}_m$ of $\boldsymbol{C}_m$. The compressor $\boldsymbol{U}_{d,m}$ is simply the first $d$ columns of $\boldsymbol{U}_m$, corresponding to the $d$ most significant eigen values in the eigen vector $\boldsymbol{s}_m$. The obtained $\boldsymbol{U}_{d,m}$ and $\boldsymbol{\mu}_m$ are then used to compress $\boldsymbol{g}_{n,m}[t + L_t], \cdots, \boldsymbol{g}_{n,m}[t + L - 1]$ of every node-$n$ in the next $L - L_t$ training iterations.

Due to the commutability, GradiVeQ gradient compression can be parallelized with RAR as shown in Algorithm 1, so that compression time can be hidden behind communication time. We place all the $N$ nodes in a logical ring, where each node can only send data to its immediate successor. We also partition every gradient vector $\boldsymbol{g}_n[t]$ into $N$ equal segments, each containing several gradient slices. The aggregation consists of two rounds: a compression round and a decompression round. To initiate, each node-$n$ will only compress the gradient slices in the $n$-th segment, and then send the compressed segment to its successor. Then, in every step of the compression round, every node will simultaneously 1) download a compressed segment from its predecessor, and 2) compress the same segment of its own. Once both are completed, it will sum the two compressed segments and send the result to its successor. After $N - 1$ steps, the compression round is completed, and every node will have a different completely aggregated compressed segment. Then in each step of the decompression round, every node will simultaneously 1) download a new compressed segment from its predecessor, and 2) decompress its last downloaded compressed segment. Note that after decompression, the compressed segment must be kept for the successor to download. The original (i.e., uncompressed) RAR is a special case of Algorithm 1, where compression and decompression operations are skipped.

**Algorithm 1** Parallelized GradiVeQ Gradient Compression and Ring All-Reduce Communication

---

1: Input: $N$ nodes, each with a local gradient vector $\boldsymbol{g}_n$, $n \in [0, N-1]$;
2: Each node-$n$ partitions its $\boldsymbol{g}_n$ into $N$ equal segments $\boldsymbol{g}_n(0), \cdots, \boldsymbol{g}_n(N-1)$;
3: Every node-$n$ compresses $\boldsymbol{g}_n(n)$ to $\boldsymbol{g}'_n(n)$ as in (3), and sends $\boldsymbol{g}'(n) \triangleq \boldsymbol{g}'_n(n)$ to node-$[n+1]_N$;
4: **for** $i = 1 : N - 1$ **do**
5:    Each node-$n$ downloads $\boldsymbol{g}'([n-i]_N)$ from node-$[n-1]_N$;
6:    **At the same time,** each node-$n$ compresses $\boldsymbol{g}_n([n-i]_N)$ to $\boldsymbol{g}'_n([n-i]_N)$ as in (3);
7:    Once node-$n$ has completed the above two steps, it adds $\boldsymbol{g}'_n([n-i]_N)$ to $\boldsymbol{g}'([n-i]_N)$, and send the updated $\boldsymbol{g}'([n-i]_N)$ to node-$[n+1]_N$;
8: **end for**
9: Each node-$n$ now has the completely aggregated compressed $\boldsymbol{g}'([n+1]_N)$;
10: **for** $i = 0 : N - 1$ **do**
11:    Each node-$n$ decompresses $\boldsymbol{g}'([n+1-i]_N)$ into $\boldsymbol{g}''([n+1-i]_N)$;
12:    **At the same time,** each node-$n$ downloads $\boldsymbol{g}'_n([n-i]_N)$ from node-$[n-1]_N$;
13: **end for**
14: All nodes now have the complete $\boldsymbol{g}''$.

---

## 3    Implementation Details of GradiVeQ

**System Overview:** At the beginning of the training, we will first apply a short warm-up phase to stabilize the model, which is common in the literature (see, e.g., [28]). Then, we will iterate between $L_t$ iterations of uncompressed aggregations and $L_c$ iterations of compressed aggregations for every gradient slice. Recall that PCA needs some initial gradient data to compute the $\boldsymbol{U}_d$ values for the linear compressor. Hence, the $L_t$ iterations of uncompressed data is used to generate the compressor which is then followed by $L_c$ iterations of compressed aggregation. Since the linear correlation drifts slowly, the compression error is curtailed by periodically re-generating the updated $\boldsymbol{U}_d$ values for the linear compressor. Hence, the interspersion of an uncompressed aggregation with compressed aggregation GradiVeQ minimizes any compression related losses.

Computing $\boldsymbol{U}_{d,m}$ from a large gradient vector is a computationally intensive task. To minimize the PCA computation overhead, GradiVeQ exploits the spatial domain consistency of gradient correlation. For every consecutive $s$ (called the compressor reuse factor) slices in the same convolutional layer, GradiVeQ computes a single PCA compressor $\boldsymbol{U}_d$ (with the slice index $m$ omitted) using the first slice, and then uses the resulting $\boldsymbol{U}_d$ to compress the remaining $s - 1$ slices. We note that, although computing $\boldsymbol{U}_d$ does not require the values from the remaining $s - 1$ slices, they should still be aggregated in an uncompressed way as the first slice, so that the parameters associated with all slices can evolve in the same way. The composition of training iterations is demonstrated in Fig. 6.

In order to reduce the bandwidth inefficiency brought by the $L_t$ iterations that are not GradiVeQ compressed, we will apply a scalar quantization technique such as QSGD [24] to these iterations, and communicate the quantized gradients.

**Parameter Selection:** We briefly describe how the various parameters used in GradiVeQ are chosen. *Slice size $K$:* We set $K$ to be a multiple of $FD$ for a convolutional layer that has $F$ filters with a depth of $D$, as the compression loss is minimized at this granularity as demonstrated in Fig. 5. In addition, $K$ should be selected to balance the compression ratio and PCA complexity. Increasing $K$ to higher multiples of $FD$ may capture more linear correlations for higher compression ratio, but will increase the computational and storage costs, as SVD is applied to a $K \times K$ covariance matrix. For example, for a $(3, 3, 16, 16)$-convolutional layer, a good choice of $K$ would be $768 = 3 \times 16 \times 16$.

*Compressor reuse factor $s$:* A larger $s$ implies that the PCA compressor computed from a single slice of size $K$ is reused across many consecutive slices of the gradient vector, but with a potentially higher compression loss. We experimented with different values of $s$ and found that the accuracy degradation of using $s = \infty$ (i.e., one compressor per layer) is less than 1% compared to the original uncompressed benchmark. Therefore, finding the compressor for a single slice in a convolutional layer is sufficient.

*Compressor dimension $d$:* the value of $d$ is determined by the maximum compression loss we can afford. This loss can be easily projected using the eigen vector $\boldsymbol{s}$. Let $0 \leqslant \lambda < 1$ be a loss threshold,

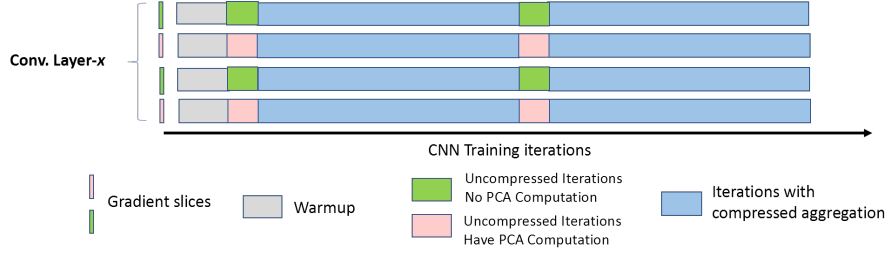

Figure 6: GradiVeQ CNN training iterations of one convolutional layer with compressor reuse factor of $s = 2$.

we find the minimum $d$ such that:

$$\frac{\sum_{k=0}^{d-1} \boldsymbol{s}[k]}{\sum_{k=0}^{K-1} \boldsymbol{s}[k]} \geqslant 1 - \lambda. \tag{6}$$

The corresponding $\boldsymbol{U}_d$ will guarantee a compression loss of at most $\lambda$ to the sample slices of size $K$ in each layer and, based on our spatial correlation observation the $\boldsymbol{U}_d$ from one slice also works well for other slices in the gradient vector.

**Number of iterations with uncompressed aggregation $L_t$:** This value could either be tuned as a hyper-parameter, or be determined with uncompressed RAR: namely perform PCA on the collected sample slices of the gradients to find out how many samples will be needed to get a stable value of $d$. We used the later approach and determined that 100 samples are sufficient, hence $L_t$ of 100 iterations is used.

*Number of compressed iterations $L_c$:* This value could also be tuned as a hyper-parameter, or be determined in the run-time by letting nodes to perform local decompression to monitor the local compression loss, which is defined as:

$$\left\| \boldsymbol{g}_n[t] - \left( \boldsymbol{U}_d \cdot \boldsymbol{g}'_n[t] + \frac{\boldsymbol{\mu}}{N} \right) \right\|^2. \tag{7}$$

When the majority of nodes experience large loss, we stop compression, and resume training with uncompressed aggregations to compute new compressors. Again, we currently use the latter approach and determine $L_c$ to be 400 iterations.

Computation Complexity: GradiVeQ has two main operations for each gradient slice: (a) one SVD over $L_t$ samples of the aggregated slice for every $L_t + L_c$ iterations, and (b) two low-dimensional matrix multiplications per iteration per node to compress/decompress the slice. Our gradient flattening and slicing (Figure 3) allows the compressor calculated from one slice to be reused by *all* the slices in the same layer, offering drastically amortized SVD complexity. On the other hand, the operations associated with (b) are of low-complexity, and can be completely hidden behind the RAR communication time (Figure 2) due to GradiVeQ's linearity. Thus, GradiVeQ will not increase the training time.

## 4 Experiments

We apply GradiVeQ to train an image classifier using ResNet-32 from the data set CIFAR-100 under a 6-node distributed computing system. The system is implemented under both a CPU cluster (local) and a GPU cluster (Google cloud). The purpose is to evaluate the gain of GradiVeQ under different levels of communication bottleneck. In the local CPU cluster, each node is equipped with 2 Ten-Core Intel Xeon Processor (E5-2630 v4), which is equivalent to 40 hardware threads. We observe that without applying any gradient compression techniques, gradient communication and aggregation overheads account for $60\%$ of the end-to-end training time, which is consistent with prior works [24]. In the GPU setup, each node has an NVIDIA Tesla K80 GPU. The increased compute capability from the GPUs magnifies the communication and aggregation overheads, which occupies $88\%$ of the end-to-end training time when no compression techniques are used.

We choose $\lambda = 0.01$ as our loss threshold. We set $s = \infty$ for each convolutional layer, which means that we only compute and use one compressor for each layer, so that the PCA overhead is

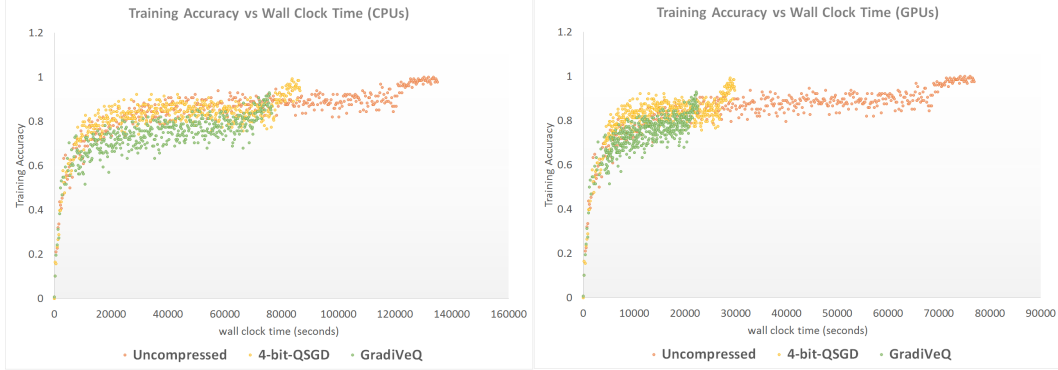

(a) For training accuracy with CPU setup, the uncompressed RAR converges around $135,000$ seconds, 4-bit-QSGD RAR converges around $90,000$ seconds, and GradiVeQ RAR converges around $76,000$ seconds. Under GPU setup, the numbers are reduce to $75,000$, $30,000$ and $24,000$, respectively.

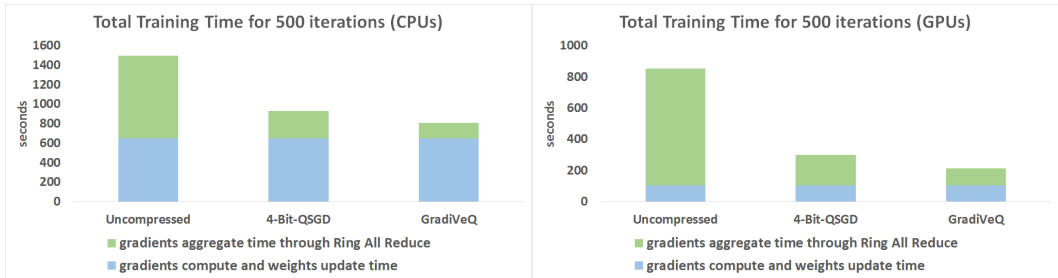

(b) End-to-End Training time breakdown for 500 iterations of three systems: Uncompressed RAR, 4-bit-QSGD RAR, and GradiVeQ RAR

minimized. Our experimental results indicate that $s = \infty$ yields no compression loss. We spend the first $2,500$ iterations on warm-up, and then periodically invest $L_t = 100$ iterations for PCA sampling, followed by $L_c = 400$ iterations of GradiVeQ-compressed gradient aggregations. With these parameter selection, we observe an average compression ratio of $K/d = 8$. The performance metrics we are interested in include wall-clock end-to-end training time and test accuracy. We compare the performance of GradiVeQ with uncompressed baseline RAR. In addition, to fairly compare GradiVeQ with scalar quantization techniques under the same compression ratio (i.e., 8), we also integrate 4-bit-QSGD with RAR. We note that for CNNs, 4-bit is the minimum quantization level that allows QSGD to gracefully converge [24]. One of the advantage of QSGD is that unlike GradiVeQ it does not need any PCA before applying the compression. GradiVeQ exploits this property of QSGD to minimize bandwidth-inefficiency during the $L_t$ iterations that are not GradiVeQ-compressed. Furthermore this approach demonstrates the compatibility of GradiVeQ with scalar quantization. We apply 4-bit-SGD to these $L_t$ iterations and use the quantized gradients for PCA.

## 4.1 Model Convergence

We analyze the model convergence of the three different systems for a given training accuracy. All the 3 systems converge after $45,000$ iterations, indicating that GradiVeQ does not incur extra iterations. However, as plotted in 7(a), both compression approaches reduce the model convergence wall-clock time significantly, with GradiVeQ slashing it even more due to its further reduction in gradient aggregation time – in our CPU setup, uncompressed RAR takes about $135,000$ seconds for the model to converge, 4-bit-QSGD takes $90,000$ seconds to converge, whereas GradiVeQ takes only $76,000$ seconds to converge. For our GPU setup in Google Cloud, these numbers are reduced to $75,000$ (uncompressed), $30,000$ (4-bit-QSGD), and $24,000$ (GradiVeQ), respectively. In terms of test accuracy, uncompressed RAR's top-1 accuracy is 0.676, while 4-bit-QSGD RAR's top-1 accuracy is 0.667, and GradiVeQ RAR's top-1 accuracy is 0.666, indicating only marginal accuracy loss due to quantization.

### 4.2 End-to-End Training Time Reduction Breakdown

The end-to-end training time consists of computation time and gradient aggregation time. The computation time includes the time it takes to compute the gradient vector through backward propagation, and to update the model parameters. The gradient aggregation time is the time it takes to aggregate the gradient vectors computed by the worker nodes. Both GradiVeQ and 4-bit-QSGD share the same computation time as the uncompressed system, as they do not alter the computations. The number is about 650 seconds per 500 ($L_t + L_c$) iterations under our CPU setup. In terms of gradient aggregation time, the uncompressed system needs 850 seconds per 500 iterations, which constitutes $60\%$ of the end-to-end training time. On the other hand, GradiVeQ substantially reduces this time by 5.25x to only 162 seconds thanks to both its gradient compression and parallelization with RAR. As a result, GradiVeQ is able to slash the end-to-end training time by $46\%$. In contrast, although 4-bit-QSGD can offer the same compression ratio, its incompatibility to be parallelized with RAR makes its gradient aggregation time almost double that of GradiVeQ. The gain of GradiVeQ becomes more significant under our GPU setup. GPUs can boost the computation speed by 6 times, which makes the communication time a more substantial bottleneck: being 88% of the end-to-end training time without compression. Thus, by slashing the gradient aggregation time, GradiVeQ achieves a higher end-to-end training time reduction of 4X over the uncompressed method and 1.40 times over 4-bit-QSGD.

## 5  Conclusion

In this paper we have proposed GradiVeQ, a novel *vector* quantization technique for CNN gradient compression. GradiVeQ enables direct aggregation of compressed gradients, so that when paired with decentralized aggregation protocols such as ring all-reduce (RAR), GradiVeQ compression can be parallelized with gradient communication. Experiments show that GradiVeQ can significantly reduce the wall-clock gradient aggregation time of RAR, and achieves better speed-up than scalar quantization techniques such as QSGD.

In the future, we will adapt GradiVeQ to other types of neural networks. We are also interested in understanding the implications of the linear correlation between gradients we have discovered, such as its usage in model reduction.

## Acknowledgement

This material is based upon work supported by Defense Advanced Research Projects Agency (DARPA) under Contract No. HR001117C0053. The views, opinions, and/or findings expressed are those of the author(s) and should not be interpreted as representing the official views or policies of the Department of Defense or the U.S. Government. This work is also supported by NSF Grants CCF-1763673, CCF-1703575, CNS-1705047, CNS-1557244, SHF-1719074.

## Footnotes

[3]GradiVeQ is not applied to the gradient of other parameters such as those from fully connected layers.

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
