[Reviews · NeurIPS 2018]

Reviewer 1



In this paper, the authors have proposed a new vector quantization technique for CNN gradient compression. GradiVeQ enables direct aggregation of compressed gradients. I have the following concerns about this paper. (1) The authors claim that existing techniques could perform poorly when used together with decentralized aggregation protocols. But they did not justify this in their paper. (2) The authors did not analyze the computational complexity of the proposed method. This should be included in the rebuttal. (3) The experiments are not convincing. The authors only employ CIFAR 100, which is not enough for this conference.

Reviewer 2



This paper proposes to boost the speed of distributed CNN training by compressing the gradient vectors before sending among the workers. The authors make an argument that the linear nature of the compression function (based on PCA) are convenient for the ring all-reduce aggregation protocal. The authors also provided empirical justification for using PCA. However, I find it hard to understand Fig 4. What exactly is plotted? What does 3 adjacent gradients mean? Do they belong to the same filter? How are they chosen? Furthermore, the authors don't seem to provide any justification for the time domain invariance. In the experiments, Fig. 7(a) on page 8 appears to show that 4-bit-QSGD achieves a higher training accuracy than GradiVeQ for the same clock time. Also, the authors used CPU rather than GPU for the experiments. Would the results be still relevant given that such neural nets train a magnitude faster with GPUs? To sum up, I do feel unconvinced that the proposed method would have a significant impact for the reasons above.

Reviewer 3



This describes how a PCA during full-vector can be used to predict a good compression method. They do a good job arguing that PCA among gradient vectors should be a common case in machine learning. The results look good. On the down side, it requires a fair amount of coding work to include it in what you do, because you still have a periodic "full gradient" phase. The PCA and how it is approximated are practical heuristics, so I don't expect a proof to be possible without a bit of fine print. I don't think there was a discussion of what happens with rare classes. In this last case, some gradients might be really badly represented. In such cases, I think you can detect the badly compressed gradient representations and "do something different". But I think the paper is OK as is, because it proposes a clean approach that can hopefully be compared with the other "do something different" compression schemes that target more the communication than the aggregation cost of distributed SGD methods. #### After authors' rebuttal and other reviews #### I thought the authors responded well to the reviewers' questions and propose some good changes to the paper. The rebuttal did not really address my class imbalance concern, which I guess also bothered Reviewer #3, who commented about the time-invariance assumption. In both cases a bothersome feature is that the applicability of a PCA assumption for gradient compression might break down, and you might not notice it. So if "GradiVeQ is able to detect compression performance by measuring the training compression loss" [rebuttal] refers to a locally measured compression fidelity, this should be in the paper --- if you can formulate a loss that degrades somewhat gracefully over the time-invariance "L" period, then both Reviewer-3 and I might be satisfied that a feature implementation has a decent chance of automagically detecting when the compressor's assumptions are beginning to malfunction. I'm keeping my overall score the same.